# Efficacy of Different Oncolytic Vaccinia Virus Strains for the Treatment of Murine Peritoneal Mesothelioma

**DOI:** 10.3390/cancers16020368

**Published:** 2024-01-15

**Authors:** Can Yurttas, Julia Beil, Susanne Berchtold, Irina Smirnow, Linus D. Kloker, Bence Sipos, Markus W. Löffler, Alfred Königsrainer, André L. Mihaljevic, Ulrich M. Lauer, Karolin Thiel

**Affiliations:** 1Department of General, Visceral and Transplant Surgery, University Hospital of Tübingen, Hoppe-Seyler-Str. 3, 72076 Tübingen, Germanyalfred.koenigsrainer@med.uni-tuebingen.de (A.K.);; 2Virotherapy Center Tübingen (VCT), Department of Medical Oncology and Pneumology, University Hospital of Tübingen, Otfried-Müller-Str. 10, 72076 Tübingen, Germany; 3Department of Internal Medicine VIII, Medical Oncology and Pneumology, University Hospital of Tübingen, Otfried-Müller-Str. 10, 72076 Tübingen, Germany; bence.sipos@med.uni-tuebingen.de; 4German Cancer Consortium (DKTK), German Cancer Research Center (DKFZ), Partner Site Tübingen, Otfried-Müller-Str. 10, 72076 Tübingen, Germany; 5BAG für Pathologie und Molekularpathologie, Rosenbergstraße 12, 70176 Stuttgart, Germany; 6Cluster of Excellence iFIT (EXC2180) “Image-Guided and Functionally Instructed Tumor Therapies”, University of Tübingen, 72076 Tübingen, Germany; 7Interfaculty Institute for Cell Biology, Department of Immunology, University of Tübingen, Auf der Morgenstelle 15, 72076 Tübingen, Germany; 8Department of Clinical Pharmacology, University Hospital Tübingen, Auf der Morgenstelle 8, 72076 Tübingen, Germany; 9Department of General, Visceral, and Thoracic Surgery, Oberschwaben Hospital Group, St Elisabethen-Klinikum, Elisabethenstr. 15, 88212 Ravensburg, Germany

**Keywords:** immunovirotherapy, oncolytic virotherapy, peritoneal carcinomatosis, intraperitoneal therapy, syngeneic murine tumor model

## Abstract

**Simple Summary:**

Effective treatment options for malignant diseases affecting the peritoneum, the inner surface of the abdominal cavity and organs, are scarce. Oncolytic virotherapy uses self-replicating viruses that target malignant cells and kill them selectively. In our work, we investigated the ability and potential of modified vaccinia viruses to infect and kill peritoneal mesothelioma cells in a mouse model. Among others, Western Reserve strain of vaccinia viruses (GLV-0b347) resulted most effective for the treatment of mouse mesothelioma. In mice diseased with mesothelioma, tumor mass and complications could be reduced and survival was prolonged when treated with vaccinia virus compared to untreated animals. Taken together, this new treatment approach showed promising results in a mouse peritoneal cancer model. Future research will focus on further improvement of this therapy.

**Abstract:**

Effective treatment options for peritoneal surface malignancies (PSMs) are scarce. Oncolytic virotherapy with recombinant vaccinia viruses might constitute a novel treatment option for PSM. We aimed to identify the most effective oncolytic vaccinia virus strain in two murine mesothelioma cell lines and the oncolytic potential in a murine model of peritoneal mesothelioma. Cell lines AB12 and AC29 were infected in vitro with vaccinia virus strains Lister (GLV-1h254), Western Reserve (GLV-0b347), and Copenhagen (GLV-4h463). The virus strain GLV-0b347 was shown most effective in vitro and was further investigated by intraperitoneal (i.p.) application to AB12 and AC29 mesothelioma-bearing mice. Feasibility, safety, and effectiveness of virotherapy were assessed by evaluating the peritoneal cancer index (PCI), virus detection in tumor tissues and ascites, virus growth curves, and comparison of overall survival. After i.p. injection of GLV-0b347, virus was detected in both tumor cells and ascites. In comparison to mock-treated mice, overall survival was significantly prolonged, ascites was less frequent and PCI values declined. However, effective treatment was only observed in animals with limited tumor burden at the time point of virus application. Nonetheless, intraperitoneal virotherapy with GLV-0b347 might constitute a novel therapeutic option for the treatment of peritoneal mesothelioma. Additional treatment modifications and combinational regimes will be investigated to further enhance treatment efficacy.

## 1. Introduction

Peritoneal mesothelioma (PM) constitutes a rare primary malignancy of the peritoneal cavity but the incidence is expected to rise due to a long latency period prior to disease onset and continued use of asbestos in developing countries. For well selected patients affected by PM, cytoreductive surgery (CRS) with hyperthermic intraperitoneal chemotherapy (HIPEC) is considered the standard of care, enabling overall survival prospects between 29 and 98 months [1,2,3,4], although randomized-controlled trials are lacking to support this treatment. However, most patients are not eligible for this aggressive therapy, limiting survival prospects to merely 6–12 months [1,2,5]. Therapeutic alternatives are therefore urgently needed in order to improve both the therapeutic efficacy and the eligibility for treatment approaches for patients affected by PM.

Oncolytic virotherapy is a novel treatment approach for malignant diseases that originated from observations in patients showing remission of their malignant disease associated with wild-type virus infections [6]. Its basic principle is that replication-competent viral vectors selectively infect and replicate within tumor cells and lyse their host cells by releasing numerous viruses as progeny [7]. Tumor selectivity is considered to be caused by depletion of interferon pathways in tumor cells, rendering them susceptible for viral infections, whilst healthy cells remain unharmed [8]. Although the mechanisms of action have not yet been fully elucidated, it is known that the virus-mediated cell lysis induces a systemic immune response that is pivotal for the success of virotherapy [9]. Meanwhile, various oncolytic viruses have already been tested in clinical trials with promising results [7,10,11] and a lead off modified herpes simplex virus has been approved in Europe for the therapy of advanced melanoma [12]. However, there is natural resistance to primary and secondary infections of oncolytic viruses limiting antitumor efficacy [11,13,14]. To overcome early interference by the immune system, one potential approach is direct virus application to the tumor site. Tumors and metastases confined to the abdominal cavity with its assumed peritoneal-plasma barrier preventing the penetration of cytostatic drugs therefore constitute an ideal target for oncolytic virotherapy [11,15,16]. Initial data from the current literature corroborate safety, feasibility, and effectiveness of this approach [11,15,17,18,19,20,21]. 

Considering the limited therapeutic options for the treatment of PM, frequent relapses even after complete cytoreductive surgery with HIPEC and the promising results of oncolytic virotherapy, our aim was to investigate the effectiveness of intraperitoneal virotherapy in this indication. In a first step, we therefore intended to assess, which of the commonly applied oncolytic vaccinia virus strains: Lister (GLV-1h254), Western Reserve (GLV-0b347), and Copenhagen (GLV-4h463), proves most effective for the treatment of PM in terms of oncolysis and virus replication. For this, the virus strains were used to infect the well-characterized murine PM cell lines AC29 and AB12 in vitro [22]. In a second step, the most effective virus strain was further investigated in two syngeneic murine models of orthotopic PM. The model was used to ensure infection of orthotopic PM by oncolytic vaccinia virus and to assess its effect concerning tumor mass reduction and overall survival in treated animals compared to controls. The results of these experiments will guide further investigations and the design of future clinical trials to assess oncolytic virotherapy as a potential new treatment option for the therapy of peritoneal surface malignancies. 

## 2. Materials and Methods

### 2.1. Cell Culture and Cell Lines

The murine PM cell lines AB12 and AC29 (kindly provided by Dr. Marc de Perrot, University of Toronto, Toronto, ON, Canada) selected for this project were generated many years ago by i.p. asbestos injection to BALB/c and CBA/j mice, respectively. Orthotopic re-administration of the cell lines to healthy syngeneic BALB/c and CBA/j mice resulted in the successful development of PM [22]. African green monkey kidney fibroblasts (CV-1 cells) were purchased from ATCC^®^ (CCL-70TM) (Manassas, VA, USA). 

Dulbecco’s Modified Eagle Medium (DMEM) supplemented with 10% fetal calf serum (FCS) (both Sigma-Aldrich, St. Louis, MO, USA) was used for cell culture. All cells were kept in culture flasks with vented caps at 37 °C with 5% CO_2_ in a humid atmosphere. Prior to use in experiments, all cell lines were tested for mycoplasma by polymerase chain reaction (PCR) using MycoSPY^®^ detection kit (M030-050; Biontex Laboratories GmbH, Munich, Germany) according to the manufacturer’s instructions.

### 2.2. Viruses

The vaccinia virus strains GLV-0b347 (Western Reserve), GLV-1h254 (Lister), and GLV-4h463 (Copenhagen) were kindly provided by Genelux Corporation (Westlake Village, CA, USA). In GLV-1h254 and GLV-4h463, the gene locus for thymidine kinase (J2R) was disrupted by insertion of a ß-galactosidase gene (*lacZ*; under control of the vaccinia early late promoter (P_7.5_)). In addition, a human transferrin receptor (*tfr*) gene was inserted in reverse orientation, downstream from P_SEL_ promoter; thus, *tfr* is not expressed. In GLV-0b347, the gene encoding for the far-red fluorescent protein TurboFP635 under control of a vaccinia synthetic early/late promoter (P_SEL_) was inserted in the same locus. F14.5L encodes for a protein responsible for virulence and cell adhesion and is the locus inactivated by insertion of a gene encoding for a green fluorescent protein (GFP) in GLV-4h463. In GLV-1h254, the *turboFP635* gene was inserted in the A56R locus, which encodes for viral hemagglutinin protein. In GLV-4h463, the A56R locus is replaced by a gene encoding for β-glucuronidase (Figure 1). For further information, see also references [23,24,25].

### 2.3. Virus Treatment, Replication, and Quantification In Vitro

For in vitro virus treatment, AB12 or AC29 cells were seeded in 24-well plates and infected on the following day with the different vaccinia virus strains at ascending multiplicities of infection (MOI) ranging from 0.0001 to 1. After one hour, the virus-containing medium was removed and replaced by fresh culture medium.

To investigate viral spread and replication of the different vaccinia virus strains, AB12 and AC29 cells were seeded in 6-well plates and infected with MOI 0.1 and 1, respectively. Samples were harvested by scraping and collecting the cells into their culture medium at 1, 24, 48, 72, and 96 h post infection (hpi). One subsequent freeze/thaw cycle was performed for cell lysis and the release of cell-bound viral particles. 

To detect infectious viral particles and to determine the virus concentration in samples, serial dilutions were titrated via plaque assay on CV-1 cells in 24-well plates as described previously [26]. After primary infection at 1 hpi, 1 mL of 1.5% (*w*/*v*) carboxymethylcellulose (Sigma-Aldrich) in DMEM with 5% (*v*/*v*) FCS, 1% (*v*/*v*) penicillin, and streptomycin was added to each well. Thereby, viral particles were confined to create plaques. Staining of viral plaques was performed at RT by using crystal violet staining solution (Carl Roth GmbH, Karlsruhe, Germany) for at least four hours to stain the remaining cell layer and visualizing the plaques. Stained virus plaques were counted and viral titers were calculated as plaque forming units per milliliter (pfu/mL). 

### 2.4. Sulforhodamine B (SRB) Cell Viability Assay

For quantification of residual cell masses after virotherapy in vitro, SRB assay was performed as described previously [27]. Treated AB12 or AC29 cells were washed with cold PBS, covered with cold trichloroacetic acid for at least 30 min for fixation, then washed four times with tap water and dried at 40 °C overnight. After staining with SRB solution (Sigma-Aldrich, Taufkirchen, Germany) for 10 min, washing with acetic acid (1%) and drying, fixed cells were dissolved in TRIS base (pH 10.5). Optical density was measured in a microtiter plate reader (Synergy HT, BioTek Instruments GmbH, Bad Friedrichshall, Germany) at a wavelength of 550 nm (reference wavelength at 620 nm). 

### 2.5. Animal Model, Interventions, and Assessment of the Peritoneal Cancer Index (PCI)

Female BALB/c and CBA/j mice were obtained from Charles River Laboratories (Sulzfeld, Germany) at the age of 4–6 weeks. Animals were kept in individually ventilated cages at a maximum of 5 individuals per cage. Prior to the interventions, mice were accustomed to the new surrounding for at least 2 weeks. Ear labeling was used for identification of individuals. Medical condition was assessed at least once daily according to a predefined score sheet. Experiments were prematurely terminated when necessary and indicated by the score sheet. 

Implantation of tumor cells to the abdominal cavity was performed under general anesthesia by i.p. injection of 0.05 mg fentanyl, 5 mg midazolam, and 0.5 mg medetomidine per kilogram bodyweight, respectively. Eyes were protected from drying by application of eye ointment (Bausch & Lomb GmbH, Berlin, Germany). Sparse shaving of the fur, disinfection, and median abdominal incision of maximum 1 cm were performed to ensure safe access to the abdominal cavity. Tumor cells (1 or 5 × 10^5^ cells of AC29 for CBA/j mice and 5 × 10^5^ or 1 × 10^6^ cells of AB12 for BALB/c mice) were diluted in 0.5 mL NaCl 0.9% and applied freely to the abdominal cavity under visual control. The abdomen was then closed by two layers of sutures. For postoperative analgesia, 20 mg carprofen and 200 mg metamizole per kg bodyweight were applied subcutaneously as well as antagonizing drugs naloxone (1.2 mg/kg bodyweight), flumazenil (0.5 mg/kg bodyweight), and atipamezole (2.5 mg/kg bodyweight). The animals were intensively observed and warmed for at least 30 min until fully regaining consciousness. Subcutaneous carprofen was reapplied if there were any signs of distress according to the score sheet. Metamizole was always accessible to the animals through the drinking water at a concentration of 1.25 mg/mL. GLV-0b347 at a dose of 5 × 10^6^ pfu diluted in 0.1 mL NaCl 0.9% was administered into the abdominal cavity by direct injection without anesthesia 5 or 10 days after tumor cell implantation as illustrated in Figure 2. For termination of the experiment, mice were killed by insufflation of CO_2_ to the cage, with subsequent subxyphoidal puncture and blood draw from the heart for virus titer analysis. Necropsy was performed by wide opening of the abdominal cavity, the thorax, and the skull in order to assess the PCI according to the method described by Ottow et al. [28] and to obtain representative samples of tumor nodules, ascites, and the brain tissue. Detection of virus and its quantification in each tissue was performed via plaque assay as described previously.

### 2.6. Statistical Analysis

All data are provided as mean ± standard deviation (SD). In vitro experiments were performed at least in duplicates. Statistical analysis and illustration were performed with GraphPad Prism version 9.4.1 software (GraphPad, LLC, San Diego, CA, USA). Welch and Brown–Forsythe versions of one-way analysis of variance (ANOVA) were applied to compare the means of remnant cell masses after virotherapeutic treatment to mock-treated cells. Level of significance is denoted by number of asterisks: *p* values ≤ 0.05 were indicated by one asterisk (*), values ≤ 0.01 with two asterisks (**), values ≤ 0.001 with three asterisks (***), and values ≤ 0.0001 by four asterisks (****) [29]. 

In order to compare PCI values, Cohen’s Weighted Kappa was calculated (SPSS Statistics Version 28, IBM, Armonk, NY, USA) [30]. The strength of agreement is denoted by the number of crosses: (^x^) indicates a poor (≤0.2) agreement, (^xx^) a fair (>0.2–0.4) agreement, (^xxx^) a moderate (>0.4–0.6) agreement, (^xxxx^) a good (>0.6–0.8) agreement, (^xxxxx^) a very good (>0.8–1.0) agreement. 

For comparison of survival by Kaplan–Meier curves, a log-rank (Mantel–Cox) test was used. Differences between treatment groups were considered statistically significant if *p* was <0.05.

## 3. Results

### 3.1. Oncolytic Effect of Vaccinia Virus Strains in Murine Peritoneal Mesothelioma Cell Lines

The in vitro treatment of murine PM cell lines AB12 and AC29 with different concentrations of oncolytic vaccinia viruses at 96 hpi showed that vaccinia virus strain GLV-0b347 (Western Reserve) was the most effective in terms of oncolysis of the three virus strains tested (Figure 3): while no relevant cell death was observed at a multiplicity of infection (MOI) of 0.1 or 1 with GLV-1h254 (Lister) and GLV-4h463 (Copenhagen) in the murine PM cell line AB12 (0% to a maximum of 20% cell death), this could be increased with GLV-0b347 (Western Reserve) to approximately 60% with an MOI of 0.1 and up to 90% with an MOI of 1. When AC29 cells were treated with GLV-4h463, oncolytic effects achieved a cell reduction by 30% at an MOI of 1 and by approximately 50% with GLV-1h254, but GLV-0b347 again proved more effective, resulting in cell reduction by 80% at an MOI of 0.1 and by >90% at an MOI of 1. 

### 3.2. Viral Replication of Vaccinia Virus Strains in Murine Peritoneal Mesothelioma Cell Lines

Analysis of viral replication after treatment of murine PM cell lines AB12 and AC29 in vitro also confirmed the higher efficacy of GLV-0b347 (Western Reserve) compared to the other viral constructs GLV-1h254 and GLV-4h463. This became evident through detecting higher viral concentrations of GLV-0b347 in both cell lines AB12 and AC29 at both MOIs 0.1 and 1, compared to the other two viral constructs (as shown in Figure 4). While at 1 hpi viral concentrations of all constructs ranged between 1 × 10^2^ and 1 × 10^4^ pfu/mL, the concentration of GLV-0b347 increased to its maximum at 48 hpi with >1 × 10^7^ pfu/mL at an MOI of 1 and >1 × 10^6^ pfu/mL at an MOI of 0.1, respectively, either plateauing and remaining stable (AB12) or decreasing slightly (AC29) thereafter. The other constructs also reached their maximum at 48 hpi, but titers remained significantly lower with a maximum of 1 × 10^6^ pfu/mL of GLV-1h254 in AC29 cells. Only viral titers of GLV-1h254 (MOI 1) in AB12 cells continued to increase after 48 hpi until peaking at 96 hpi, but remained below the value of GLV-0b347 at an MOI of 1 with ~1 × 10^6^ pfu/mL.

### 3.3. Intraperitoneal Virus Treatment of Peritoneal Mesothelioma in a Syngeneic Murine Model

Initially, 5 × 10^5^ AC29 cells for each of 20 CBA/j mice and 1 × 10^6^ AB12 cells for each of 20 BALB/c mice (aged 6–8 weeks) were applied to anesthetized animals by laparotomy (Figure 5A). Abdominal wall closure was performed with braided sutures in two layers (Figure 5B). Oncolytic virus GLV-0b347 or saline in control animals was injected i.p. ten days after tumor implantation to groups of ten AC29-bearing CBA/j mice or ten AB12-bearing BALB/c mice, respectively. At the termination of the experiment (24 and 48 hpi, respectively), the abdominal cavity was thoroughly inspected for tumor tissue and concomitant signs like ascites and intestinal obstruction. Tumor burden was quantified (Figure 5C) and representative samples were taken for further examinations. 

The mock-treated arm in the AB12-induced PM mouse model showed a mean value for the peritoneal cancer index (PCI) of 1.2 at 24 hpi versus 1.0 at 48 hpi. After virus application to the abdominal cavity, the mean PCI value was 0.8 at 24 hpi and 1.0 at 48 hpi. In the PM mouse model using AC29 tumor cells, a mean PCI value of 2.2 and 2.0 at 24 hpi and 48 hpi were assessed, respectively. After virus treatment for 24 h, the mean PCI value was 1.8 and after 48 h, the respective value was 1.2. 

Representative tumor lesions from mock as well as virus-treated animals were obtained and assessed by plaque assay to detect and quantify the oncolytic virus GLV-0b347. Live and infectious virus was detected in i.p. tumors of three of five AB12-bearing mice (60%) and two of five mice with intraabdominal AC29 tumor cells (40%). Results are illustrated in Figure 6A,B. 

In the next step, i.p. virus was applied again 10 days after open tumor cell implantation. The animals in groups of five animals each were then observed for therapy response and evaluation of virus replication after 7 or 14 days compared to control animals, which were treated with saline. Due to rapid tumor progression, the endpoints specified in the score sheet were reached prematurely in most animals, so that the experiment could not be carried out as planned. Even though the virus-treated (green lines) AB12 tumor-bearing BALB/c mice lived on average one day longer (Figure 7A) and the AC29 tumor cells harboring CBA/j mice half a day longer (Figure 7B) as compared to mock-treated animals (black lines), there was no statistically significant (ns) difference concerning the survival rate of the animals in this experimental setup (*p* = 0.2322 and *p* = 0.2589, respectively).

The necropsy of the animals accordingly showed a pronounced abdominal tumor load and development of bloody ascites in AC29 tumor-bearing CBA/j mice. The most probable cause for the rapid deterioration of the condition of most of the test animals was an obstruction in the upper gastrointestinal tract with gastric outlet stenosis as well as massive ascites production.

Therefore, reduction in the amount of tumor cells applied to the abdominal cavity (1 × 10^5^ instead of 5 × 10^5^ AC29 tumor cells in CBA/j mice and application of 5 × 10^5^ instead of 1 × 10^6^ AB12 cells in BALB/c mice) and earlier onset of virotherapeutic treatment after tumor cell implantation (five instead of ten days) were conducted in order to initiate treatment already at a less advanced stage of disease. With these modified experimental conditions, each group of AC29 (*n* = 10) and AB12 (*n* = 10) tumor-bearing mice was either treated by i.p. application of GLV-0b347 or with i.p. saline application as mock control, respectively, and were subsequently observed for a maximum of 14 days after virus application or until they showed signs of distress according to a predefined score sheet. Survival of CBA/j and BALB/c mice that were treated by GLV-0b347 was significantly prolonged (*p* = 0.094 and 0.0004, respectively, depicted in Figure 8A,B), when compared to mock-treated animals. 

Necropsy of virus-treated animals showed lower PCI values in AB12 and AC29 tumor-bearing animals when compared to mock-treated animals (Figure 9A,B and Figure 10A–D). Representative findings from necropsy are illustrated in Figure 10. Tumor involvement of the visceral and parietal peritoneum was moderate in mock-treated AB12 cells in BALB/c mice (Figure 10A) and excessive in mock-treated AC29 cells in CBA/j mice (Figure 10C). After i.p. treatment with GLV-0b347, tumor amounts were reduced so that there were no or almost no tumor deposits visible (Figure 10B,D). PCI was on average 2.7 in mock-treated AB12 mice compared to 1.8 in virus-treated mice (Cohen’s Kappa = 0.151 indicating a poor agreement). In AC29 mice, mean PCI was 2.7 without virus treatment, whereas in mice treated with i.p. GLV-0b347, PCI was on average 1.6 (Cohen’s Kappa = 0.22 indicating a fair agreement). Ascites was less frequently observed in AC29 tumor-bearing animals (20%) that received GLV-0b347, while all mock-treated animals showed ascites (100%) (Figure 9C). There was no occurrence of ascites in AB12 carrying BALB/c mice, irrespective of mock or virus treatment. 

## 4. Discussion

Our objective was to identify the most effective among commonly applied oncolytic vaccinia virus strains in two cell lines of murine PM and to assess the efficacy of this strain for the treatment of PM in a murine syngeneic orthotopic model. GLV-0b347 (Western Reserve) was most effective in terms of oncolysis and viral replication in AB12 and AC29 cell lines of murine PM in vitro. In vivo, the virus construct was shown to infect and replicate within PM cells. Whereas GLV-0b347 was without benefit for animals with advanced tumor growth, PCI and ascites production were reduced and overall survival significantly prolonged in animals with modified treatment conditions (i.e., reduced tumor burden and earlier application of intraabdominal therapy) that were treated by single i.p. application of GLV-0b347 compared to mock-treated animals. 

A comparable animal experiment investigating the same PM cell lines in a syngeneic mouse model has already been published in 2014 [15]. In this study, a genetically modified vaccinia virus of the Western Reserve strain with deletions of thymidine kinase and viral growth factor genes and insertion of a red fluorescent marker gene was used at single doses of 10^9^ pfu/mL. The therapy was demonstrated to be effective with cases of animals considered cured by the treatment. As observed in our experiments, the lower the tumor burden at initiation of therapy, the more pronounced were the resulting effects, with virotherapy proving most effective when simultaneously applied to the abdominal cavity at tumor cell implantation, resulting in effects comparable to complete surgical cytoreduction. It is however noteworthy that the virus amounts applied in the work by Acuna et al. were 1 × 10^9^ pfu/mL in contrast to only 5 × 10^6^ pfu/mL GLV-0b347 used during our experiments. Despite the impressive results reported by Acuna et al., no further studies or clinical trials have since investigated this approach following the publication of these promising results. 

The significance of the intraabdominal tumor extent for eligibility for cytoreductive surgery (CRS) and HIPEC as well outcomes after treatment has been demonstrated by numerous studies [31,32,33,34,35]. Excessive tumor loads might therefore also explain the limited therapeutic efficacy evidenced in clinical trials so far, when investigating i.p. application of oncolytic virotherapy to patients with late-stage peritoneal carcinomatosis [11,36,37]. Hence, i.p. virotherapy might prove more effective at early disease stages with limited peritoneal tumor dissemination or after surgical cytoreduction, although this assumption remains hypothetical so far and requires confirmation by future trials. In patients treated with CRS and HIPEC for peritoneal surface malignancies, complete or almost complete surgical cytoreduction is established as a crucial prognostic factor in order to prolong progression-free and overall survival [2,32,38]. In the context of CRS, HIPEC is believed to be required for eliminating residual tumor cells within the abdominal cavity that potentially remain after surgery. Although HIPEC following CRS has been shown effective for prolonging overall survival in patients with ovarian cancer [39], pseudomyxoma peritonei [40] and PM [41], such benefits are mostly limited. Some investigations have even demonstrated the futility of HIPEC [42,43,44]. Thus, the significance of HIPEC treatment in patients with peritoneal surface metastases is heavily debated and there is currently an urgent unmet medical need for patients that undergo CRS for additional treatments. 

Against this background, the selectivity for cancer cells as well as the self-amplifying nature of oncolytic virotherapy make it an appealing approach as an additive treatment following CRS. 

Apart from the use as an auxiliary treatment approach for CRS, the efficacy of i.p. virotherapy may be enhanced by modifications of dose, frequency, and timing of virus application as well as combination treatments with, e.g., systemically applied chemotherapy or immune checkpoint inhibitors (ICIs). Ishikawa et al. assessed the oncolytic potency of an attenuated adenovirus applied to peritoneal metastasis from gastric cancer in combination with paclitaxel. Synergistic treatment effects were demonstrated for in vitro treatment of gastric cancer cell lines as well as in vivo using an orthotopic peritoneal cancer mouse model from gastric cancer showing both enhanced anticancer efficacy as well as viral replication in tumor cells [20]. In 2018, Kowalsky et al. investigated the combined use of an oncolytic vaccinia virus (Western Reserve strain) with an anti-PD-1 antibody in immunocompetent mouse models of colon and ovarian cancer. The results showed enhanced tumor regression and prolonged survival of the experimental animals receiving a combined treatment compared to the respective monotherapies [45]. A similar study combining the modified herpes simplex virus talimogene laherparepvec (T-VEC) with CTLA-4 blockade was conducted in a melanoma mouse model, for which an increased efficacy of the combination therapy was reported by the authors [46]. Furthermore, results from a clinical trial in patients with metastatic malignant melanoma treated by oncolytic virotherapy together with ICIs were published in 2017 by Ribas et al. [10]. The virotherapeutic T-VEC (based on a herpes simplex virus type-1) was administered intratumorally combined with i.v. anti-PD-1 ICI (pembrolizumab) in patients with metastatic melanoma. This combination therapy showed an overall response rate of 62%, a complete response rate of 33%, and found evidence for improved CD8+ T cell infiltration through T-VEC administration into tumors. Similar results were obtained in a phase II trial when investigating combined T-VEC with ipilimumab vs. ipilimumab alone for the treatment of advanced melanoma [47]. In this indication, the objective response rates were significantly higher when combining ICI therapy with T-VEC and respective outcomes proved durable. Based on such results, it may be assumed that oncolytic virotherapy is indeed able to increase the efficacy of ICI treatment. Further combinatory treatment approaches involve arming oncolytic viruses to transfer so-called suicide genes encoding for enzymes that can locally toxify systemically applied prodrugs [17]. Kurosaki et al. investigated a recombinant armed vaccinia virus expressing cytosine deaminase and uracil phosphoribosyl transferase to convert the prodrug 5-fluorocytosine to cytostatic 5-fluorouracil, testing the approach in pancreatic cancer cell lines and mouse models of peritoneal or hepatic metastasis. The employed vaccinia virus was able to selectively infect pancreatic cancer cells, whilst sparing benign stromal tissue and oncolytic effects could be enhanced by the combination with the prodrug in vitro and in vivo [17]. Respective clinical trials have not been conducted yet, but hold promising potential. 

Prior to performing clinical trials, however, further in vitro experiments to increase the understanding of oncolytic virotherapy and its interactions with the immune system are essential. Assays using organoids derived from peritoneal surface malignancies obtained during surgery combined with autologous peripheral-blood mononuclear cells may help to investigate the role of the immune system. The significance of the results presented here is certainly limited by the highly artificial model of murine PM, so that our findings are not necessarily directly transferable to the treatment of patients with PM. When comparing results of oncolytic vaccinia virus treatment from human pleural mesothelioma cell lines in vitro and in vivo [48] to our results, the murine PM cell lines seem to be less sensitive to this treatment. Even when compared to each other, the two murine tumor models of PM, although initially induced by the same technique of applying asbestos into the peritoneal cavity of BALB/c and CBA/j mice [15], show different malignant characteristics: AB12 tumor cell-bearing BALB/c mice required comparably more tumor cells in order to develop diffuse peritoneal tumor dissemination than CBA/j mice did with AC29 tumor cells, indicating a more aggressive behavior of AC29 tumor cells. Ascites was evidenced only in the AC29 mesothelioma model in CBA/j mice but not with AB12 tumor cells in BALB/c mice. 

Although oncolytic virotherapy, when applied to the abdominal cavity for the treatment of peritoneal surface malignancies has already been demonstrated to be feasible and safe [11,36,37], there are factors limiting efficacy. It stands to reason that multimodal treatment, ideally at an early disease stage or at least after downstaging through pretreatment or surgery, will likely improve results of oncolytic virotherapy for the therapy of peritoneal surface malignancies. Prior to investigating potential combination therapies in clinical trials, pre-clinical evidence is required to support the most promising therapeutic approach. With our investigations, we were able to demonstrate that oncolytic vaccinia virus GLV-0b347 replicated within murine PM cells lines in vitro and effectively lysed them, performing better than the oncolytic vaccinia viruses GLV-4h463 and GLV-1h254. It is so far unclear, why the virus constructs behaved differently in terms of oncolysis and which are the underlying factors. One possible explanation could be the number of inserted marker genes. Whereas GLV-0b347 constituting the most effective construct for the treatment of murine PM was modified by one additional marker gene, GLV-1h254 and GLV-4h463 contain two and three additional marker genes, respectively. It can therefore be speculated that the additional genetic load may suppress the virulence and oncolytic potency of virotherapeutic constructs. In vitro analyses showed previously that oncolytic vaccinia constructs were increasingly attenuated the more additional genes were inserted [49]. 

## 5. Conclusions

In the investigated models of murine PM in the here presented work, i.p. applied GLV-0b347 was able to significantly prolong overall survival, decrease the intraabdominal tumor load, and inhibit the formation of ascites. However, effective treatment was only observed in those animals with limited tumor burden at the time point of virus application. We were therefore not only able to demonstrate the effects of intraperitoneal virotherapy with GLV-0b347 with therapeutic potential for future treatment of PM, but also became aware of its dependence on limited peritoneal tumor burden when beginning the treatment. Future research will investigate combinational treatment approaches to further enhance efficacy. So far, it is unclear whether our results are transferable to the treatment of humans with PM, which has to be clarified by further research and ultimately in clinical trials.

## Figures and Tables

**Figure 1 cancers-16-00368-f001:**
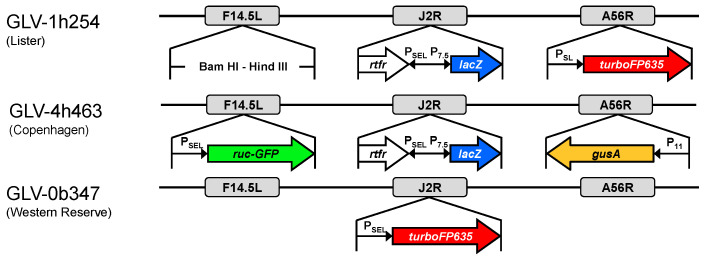
Different strains of oncolytic vaccinia virus; (**top**): GLV-1h254 (Lister), (**middle**): GLV-4h463 (Copenhagen), and (**bottom**): GLV-0b347 (Western Reserve).

**Figure 2 cancers-16-00368-f002:**
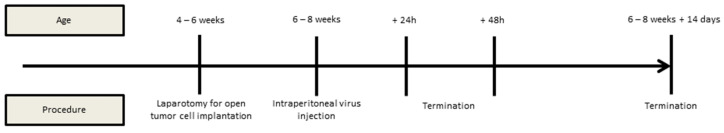
Schematic illustration of the timing of treatment procedures in the in vivo experiments. Infection of tumors by intraperitoneal virus application was assessed by experiments that were terminated 24 h and 48 h post infection, respectively. Overall survival was investigated over 14 days after intraperitoneal virus application.

**Figure 3 cancers-16-00368-f003:**
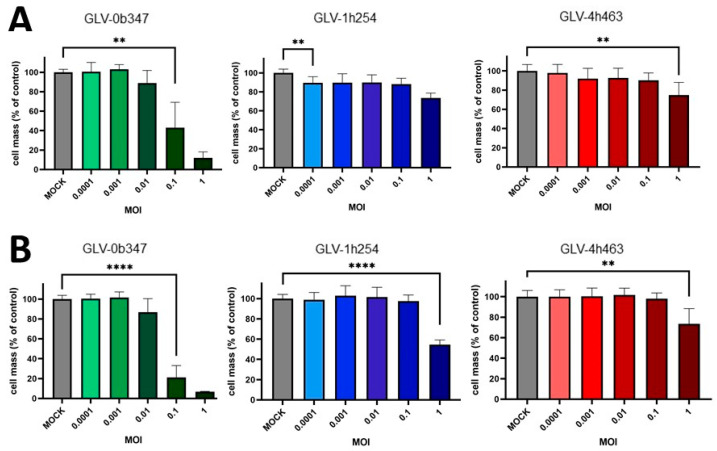
Viability of murine peritoneal mesothelioma cell lines after treatment with different vaccinia virus strains. AB12 (**A**) and AC29 (**B**) tumor cells were infected with GLV-0b347 (Western Reserve), GLV-1h254 (Lister) or GLV-4h463 (Copenhagen) at various multiplicities of infection (MOIs) ranging from 0.0001 to 1 or remained uninfected (mock). At 96 h post infection (hpi), the remaining tumor cell masses were determined by SRB viability assay. Vaccinia virus mediated oncolysis was calculated relative to mock control. The mean ± SD of at least two independent experiments performed in quadruplicates is shown. For clarity reasons, only the lowest MOI that first reached statistical significance is annotated by asterisks. *p* ≤ 0.01 (**), *p* ≤ 0.0001 (****).

**Figure 4 cancers-16-00368-f004:**
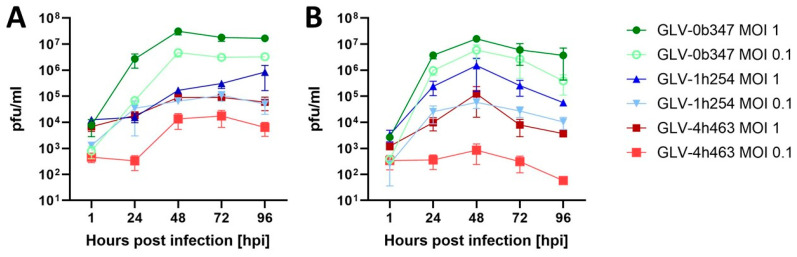
Viral replication of different vaccinia virus strains in murine peritoneal mesothelioma cell lines. AB12 (**A**) and AC29 (**B**) tumor cells were infected with GLV-0b347 (Western Reserve), GLV-1h254 (Lister) or GLV-4h463 (Copenhagen) at indicated MOIs and viral replication was analyzed via plaque assay at 1, 24, 48, 72, and 96 hpi. The mean ± SD of at least two independent experiments performed in duplicates is shown. hpi: hours post infection; MOI: multiplicity of infection; pfu: plaque-forming units.

**Figure 5 cancers-16-00368-f005:**
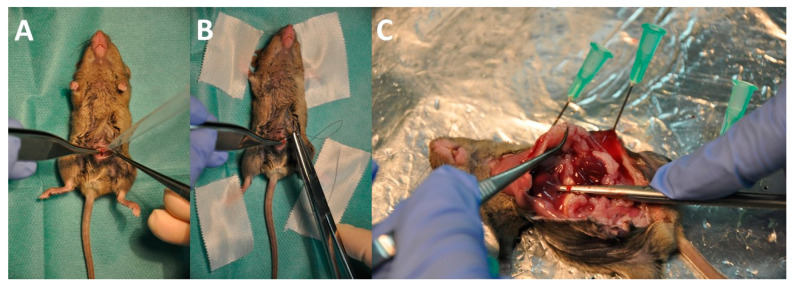
Intraperitoneal treatment of peritoneal mesothelioma in a murine tumor model of AC29 tumor cells in CBA/j mice with GLV-0b347. (**A**) Open implantation of AC29 cells into the peritoneal cavity of anesthetized CBA/j mouse. (**B**) Two-layered abdominal wall closure with braided sutures. (**C**) Exploration of the abdominal cavity after euthanasia 48 h after intraabdominal virus application showing multiple indured tumor formations. The tweezers point to a subphrenic mass of peritoneal mesothelioma beneath the left diaphragm.

**Figure 6 cancers-16-00368-f006:**
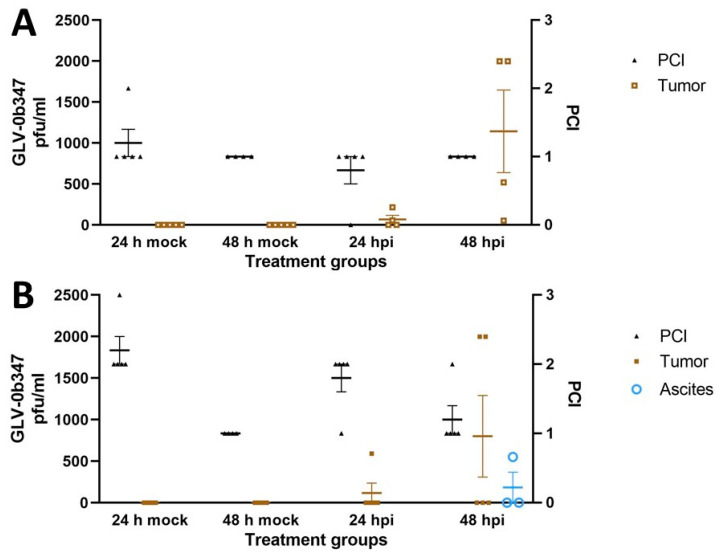
Detection of infectious GLV-0b347 viral particles (pfu/mL) in tumor lysates 24 and 48 h after treatment of i.p. tumors in AB12 tumor cell-bearing BALB/c mice (*n* = 5) (**A**) and AC29 tumor cells in CBA/j mice (*n* = 5) (**B**), respectively, and extent of peritoneal tumors quantified by PCI value according to Ottow et al. [28]. hpi: hours post infection; i.p.: intraperitoneal; pfu: plaque-forming unit; PCI: peritoneal-cancer index.

**Figure 7 cancers-16-00368-f007:**
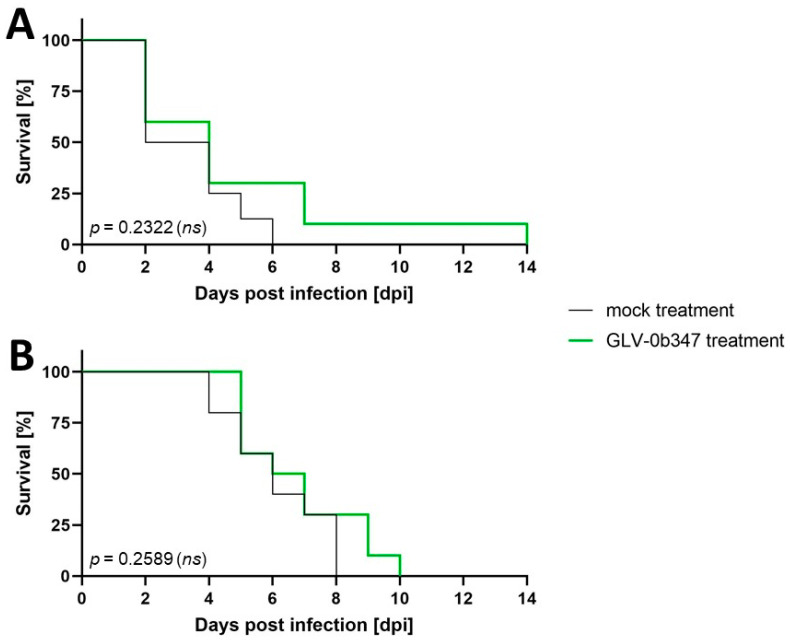
Survival of AB12 tumor-bearing BALB/c mice (*n* = 10) (**A**) and AC29 tumor-bearing CBA/j mice (*n* = 10) (**B**) in the course of 14 days after virus treatment with GLV-0b347 (green) and in control (mock; black), respectively. There was no statistically significant (ns) difference regarding survival between treatment groups. dpi: days post infection; ns: not significant.

**Figure 8 cancers-16-00368-f008:**
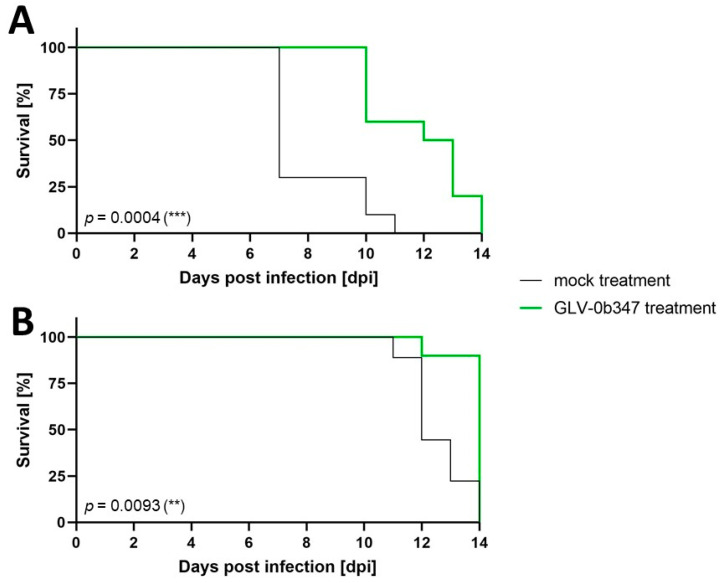
Kaplan–Meier survival curves of AB12 (*n* = 10) (**A**) or AC29 (*n* = 10) (**B**) tumor-bearing mice treated with modified experimental conditions including surgical implantation of a reduced tumor cell number and earlier onset of i.p. virotherapy with GLV-0b347 (green) compared to control animals (mock; black) until 14 dpi. dpi: days post infection. *p* < 0.01 (**), *p* < 0.001 (***).

**Figure 9 cancers-16-00368-f009:**
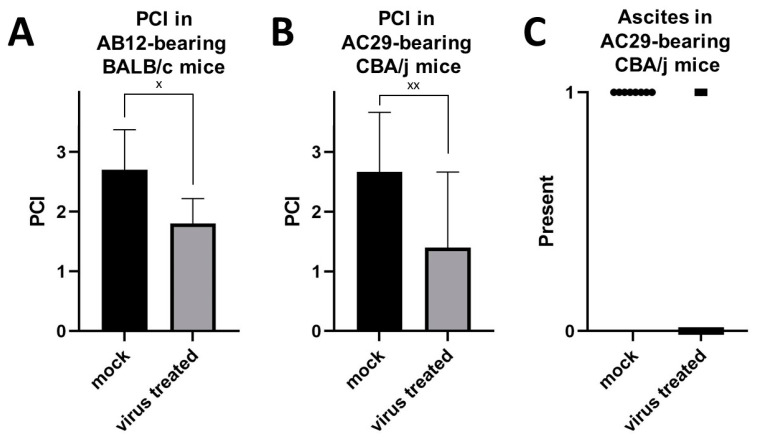
Intra-abdominal tumor burden given as peritoneal cancer index (PCI) quantified according to Ottow et al. [28] at individual end of experiment based on either signs of strain or maximum 14 days after virus application in AB12 (*n* = 10) (**A**) or AC29 (*n* = 10) (**B**) tumor-bearing mice treated with GLV-0b347 i.p. compared to control animals (mock). (**C**) Presence of ascites in AC29 mice (*n* = 10) treated with GLV-0b347 compared to control animals (mock). Each symbol indicates one mouse. The strength of agreement is given by the number of crosses: (^x^) indicates poor (≤0.2) agreement, (^xx^) fair (>0.2–0.4) agreement.

**Figure 10 cancers-16-00368-f010:**
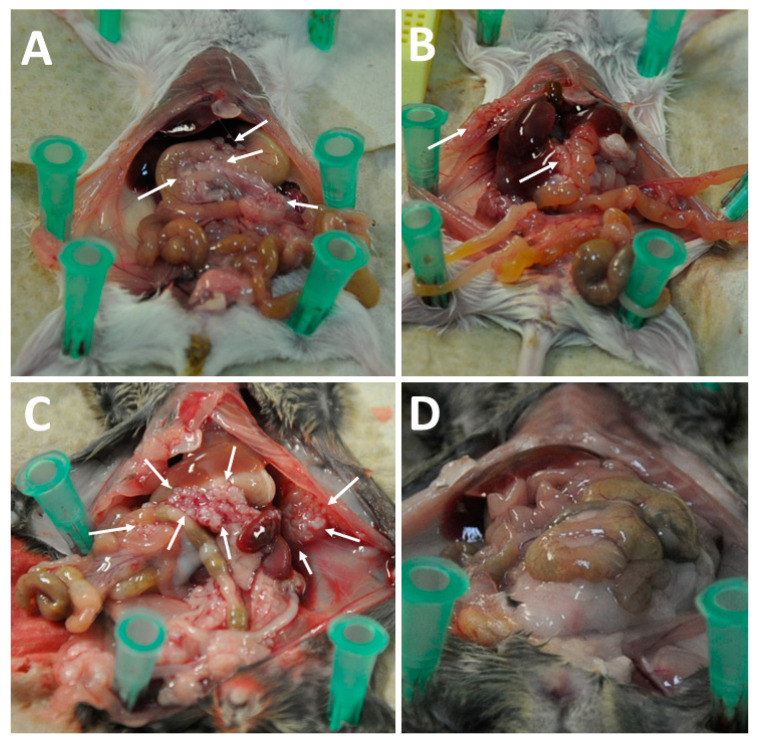
AB12 tumor cells in BALB/c mice after mock treatment show tumor deposits (indicated by white arrows) at the parietal and visceral peritoneum (**A**) compared to GLV-0b347-treated animals with hardly any tumor left over (**B**) at the time of necropsy. (**C**,**D**) Depict AC29 cells in CBA/j mock-treated mice with excessive tumor load ((**C**), white arrows) compared to virus-treated mice without any signs of residual tumors (**D**).

## Data Availability

Data are available from the corresponding author upon personal request.

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
