# Peer review of "Efficacy of Different Oncolytic Vaccinia Virus Strains for the Treatment of Murine Peritoneal Mesothelioma"

_cancers, 2024, doi:10.3390/cancers16020368_

Round 1
Reviewer 1 Report
Comments and Suggestions for Authors
· In general, there are some typo errors and grammar mistakes. Please, make sure about them.
· Please, rewrite the main finding in abstract and conclusion sections to be clear and readable for readers in future. (Do focus on novelty of your work) highly recommended.
· Please, do add key words that are more specific.
· The Introduction section need to improve. In introduction, you have to write sufficient background information, and the purpose of the article is clearly defined at the end of the introduction.
· Please do cite the following articles in missing places especially in materials and methods section.
* Pease do cite these articles as a reference for statistical analysis.
Ali, I. H., Jabir, M. S., Al-Shmgani, H. S., Sulaiman, G. M., & Sadoon, A. H. (2018, May). Pathological And immunological study on infection with escherichia coli in ale balb/c mice. In Journal of Physics: Conference Series (Vol. 1003, No. 1, p. 012009). IOP Publishing.).
· Please add the conclusion section as a separate section. · Please, make sure about the resolution of some images, and figures.
Comments on the Quality of English Language
Please see my comments for authors.
Reviewer 2 Report
Comments and Suggestions for Authors
The work by Yurttas et al explores the application of oncolytic vaccinia virus strains for their ability to replicate in and destroy murine mesothelioma cells in vitro and in vivo. While the work is sound a properly interpreted, it is quite preliminary. Additional assays would strengthen the work, but I understand the investment that such assays require. I do not wish to impede the dissemination of these results.
It seems that the murine cell lines are poor responsders to virotherapy. Reports in the literature (ex.,doi: 10.1016/j.omto.2020.08.011) used human cell lines that seemed to be quite a bit more sensitive to treatment. The authors may wish to discuss the particular aspects of the murine model that may contribute to the limited response. Is it known that human vs. mouse cell lines respond differently to vaccinia? Are antiviral immune responses expected in the immune competent mouse model?
I suggest that the authors add a schematic representation of the in vivo assays to more clearly illustrate the treatment/euthanasia regimen. This was a bit hard to grasp from the text.
The work may be strengthened with additional assays, such as spheroid models, PBMC infiltration in spheroid, examination of splenocytes to explore immune activation. I would imagine that such assays will be addressed in the future.
Reviewer 3 Report
Comments and Suggestions for Authors
The manuscript entilted "Efficacy of different oncolytic vaccinia virus strains for the treatment of murine peritoneal mesothelioma" attempts to assses effiacy in mesothelioma treatment by onolytic virus in in vitro and murine in vivo models. Althought the concept is not new, authors showed interesting dependency between cancer load and treatment effect. From my point of view this is the most imporatnt conclusion from all the work which can be helpful to further develope this treatment approach. Neverthenless I have questions and comments for authors:
1. What was specific procedure for cancer implementation. It is written that cellse were implemented into abdomen under visual control, but were there injected into peritoneal tissue or freely into abdominal cavity?
2. I suggest to describe more detailed the mechanism of action of oncolytic viruses (even is not yet fully elucidated) and explain (if there is available literature) mechanism of its specific tropism to cancers and not to healthy cells.
3. Are there available any informations that could explain difference in repsonse to virus treatment by two different cell lines? Are there known differences in mutations in suppresor genes or overexpression of oncogenes and therefor could suggest affected signaling pathway?
4. Could be with great value to include pictures from necropsy where tumor nodules are visible and maybe some visual differences between mock and treated nodules could be seen.
